# Becoming more of an insider: A grounded theory study on patients' experience of a person-centred e-health intervention

Emmelie Barenfeld[1,2]*, Lilas Ali[1,2], Sara Wallström[1,2], Andreas Fors[1,2,3], Inger Ekman[1,2]

1 Institute of Health and Care Sciences, Sahlgrenska Academy, University of Gothenburg, Gothenburg, Sweden, 2 University of Gothenburg Centre for Person-Centred Care (GPCC), Sahlgrenska Academy, University of Gothenburg, Gothenburg, Sweden, 3 Närhälsan Research and Development Primary Health Care, Region Västra Götaland, Sweden

* emmelie.barenfeld@gu.se

**Data Availability Statement:** The study was approved by the Regional Ethical Review Board in Gothenburg. As dictated by the ethical body that approved the study and the promise to participants

## Abstract

### Objective

The aim was to explore the experiences of a person-centred e-health intervention, in patients diagnosed with chronic obstructive pulmonary disease (COPD) or chronic heart failure (CHF).

### Design

Grounded theory was applied to gather and analyse data.

### Setting

The study is part of a research project evaluating the effects of person-centred care (PCC) using a digital platform and structured telephone support for people with COPD or CHF recruited from nine primary care units in Sweden.

### Participants

Twelve patients from the intervention group were purposefully selected in accordance with the initial sampling criteria.

### Intervention

The intervention was delivered through a digital platform and telephone support system for 6 months. The intervention relied on person-centred ethics operationalised through three core PCC components: patient narratives, partnership and shared documentation.

### Results

A core category was formulated: *Being welcomed through the side door when lacking the front door keys*. The core category reflects how a PCC intervention delivered remotely provides access to mutual and informal meetings at times when professional contacts were

in their informed consent, the raw study data cannot be shared publicly as the data contain potentially identifying or sensitive patient information. Data will be stored for 10 years at the University of Gothenburg to enable review. Data are available for researchers who meet the criteria for access to confidential data. Data is covered by the Public Access to Information and Secrecy act and a confidentiality assessment will be performed at each individual request. Permission from University of Gothenburg, the Institute of Health and Care Science, has to be obtained before data can be accessed. Access could be obtained by contacting Swedish National Data service (SND), University of Gothenburg, Box 463, 405 30 Gothenburg, Sweden. Tel. +46 31-786 10 00. E-mail: snd@gu.se. Dataset DOI: https://doi.org/10.5878/8ycn-k945.

**Funding:** The Swedish Heart & Lung Foundation (DNr.20180183). https://www.hjart-lungfonden.se/om-oss/in-english/ Main-applicant IE. The Swedish Research Council (DNr 2017-01230). https://www.vr.se/english.html Main-applicant IE. The Centre for Person-Centred Care at the University of Gothenburg (GPCC), Sweden. https://www.gu.se/en/gpcc Main-applicant IE. Hjalmar Svensson foundation, Sweden (HJSV2020070) Main-applicant EB. The funders had no role in the study design, data collection and analysis, decision to publish, or preparation of the manuscript.

**Competing interests:** The authors have declared that no competing interests exist.

desired to support patient self-management goals. According to patients' wishes, family and friends were seldom invited as care partners in the e-health context.

## Conclusions

A PCC intervention delivered remotely as a complement to standard care in a primary care setting for patients diagnosed with COPD or CHF is a viable approach to increase patients' access and involvement in preventive care. The e-health intervention seemed to facilitate PCC, strengthen patients' position in the health service system and support their self-management.

## Introduction

Visions of future healthcare have identified the need for both person-centred care (PCC) and for the digitalisation of health services [1–3], which has the potential to enhance equal access to care and to improve the effectiveness, accessibility and safety of health care in patients with chronic conditions. Chronic obstructive pulmonary disease (COPD) and chronic heart failure (CHF) are two such conditions that negatively affect quality of life [4] and the ability to perform daily activities [5,6]. Looking to the future, there is a need to develop and evaluate person-centred e-health interventions to support individuals in managing their symptoms and everyday life.

In general, e-health services (including telehealth and digital platforms) provide a promising strategy for strengthening preventive measures and self-management in patients with chronic conditions [7]. Additionally, these services support access to health information and facilitate communication. Historically, e-health interventions have relied on one-way communication of health information from health professional to the patient [8,9]. Even where there has been a shift towards more interactive service, this has modelled conventional roles, with the health professional in the role of expert providing monitoring, assessments or education, and patient participation limited to the reporting of signs or symptoms [7]. In contrast, a person-centred approach considers both patients and professionals to be experts, emphasising the benefits of working in partnership to enhance health and wellbeing, and at the same time promoting the self-efficacy needed for self-care [10]. A starting point in PCC is to listen to and to help the patient identify resources for managing symptoms and everyday life events, through transparent dialogue and shared documentation of a health plan. The implementation of such actions, essential to PCC, is influenced by the intervention content and design and by the care environment [11,12]. In the context of e-health, research capturing the patient experience is needed to understand why particular e-health interventions do or do not achieve the desired outcomes [13]. To support the future development and realisation of PCC in the e-health context, this study investigated patients' experience of one such intervention—the 'person-centred care at distance'(PROTECT) intervention [14].

The PROTECT intervention [14] combines the use of a digital platform and structured telephone support, building on previous evidence of health improvements from e-health-based PCC [15,16]. The intervention aims to enhance self-efficacy, a critical personal resource in self-management and the improvement of health outcomes [17]. A few studies have reported on the use of e-health interventions—including remote monitoring and telecare—to support self-management, from the perspective of individuals living with COPD or CHF [13,18,19]; however, none of these explored a person-centred e-health intervention. Therefore, little is

known about how intervention content and design contribute to the patient experience of PCC in an e-health context. In addition, studies exploring patients' perspectives on the benefits and challenges of PCC in an e-health context are scarce. *Therefore, this study aimed to explore experiences of a person-centred e-health intervention, in patients diagnosed with COPD or CHF.*

## Materials and methods

A grounded theory approach inspired by Charmaz [20] was applied to guide the study design. With its focus on exploration of social processes and contextual influences, grounded theory is suitable to deepen our understanding of actions and circumstances contributing to PCC in an e-health setting. We followed the Standards for Reporting Qualitative Research reporting guidelines [21].

### Study setting and the PROTECT intervention

This study is one element of a larger research project, the PROTECT project (NCT03183817) [14], designed to evaluate a PCC intervention delivered remotely as an adjunct to usual care. Participants of the PROTECT trial, a randomised controlled trial, were patients diagnosed with CHF or COPD recruited from nine urban public primary care centres in Sweden. The PROTECT intervention consisted of one or more person-centred telephone conversations with a dedicated health professional (from nursing, physiotherapy, medicine and/or occupational therapy) trained in various communication skills (e.g., listening, open-ended questions, reflections and summaries). Patients were offered an unlimited number of phone calls, and the number was determined based on an agreement between the patient and health professional. In addition to the telephone support, the intervention included access to an interactive digital platform. If desired by the patient, family and friends of the patient were invited to access the digital platform. Table 1 shows the possible uses of the digital platform among the patients, professionals, family and friends. Further details on the intervention are provided elsewhere [14].

The theoretical starting point was grounded in person-centred ethics [22] and operationalised as described by Ekman et al. [10]. In their operationalisation, three fundamental

**Table 1. An overview of the content in the PROTECT intervention, expected use and access to digital platform functions in the three user groups.**

| Platform function | Expected use | User access | | |
|---|---|---|---|---|
| | | **Patient** | **Professionals** | **Family and friends** |
| **Write and receive messages** | Optional | Yes | Yes | Yes* |
| **Direct dial to the professional team** | Available during office hours | Yes | Yes | Yes* |
| **Self-ratings** | Optional degree of use | Report | - | - |
| | | Monitor | Monitor | Monitor* |
| | | Follow trend graphs | Follow trend graphs | Follow trend graphs* |
| **Personal notes** | Optional degree of use | Write | No | No |
| | | Read | | |
| **Health plan** | Updated after each phone call | Write | Write | - |
| | | Overview | Overview | Overview* |
| | | Agree | Agree | - |
| **Invitation of family and friends** | Optional choice | Yes | No | No |
| **View links to health information and supportive networks** | Optional degree of use | Yes | Yes | Yes* |

*If decided by the patient.

principles to facilitate and safeguard PCC outlined a partnership between health professionals and patients (including family and friends when requested) [10]: First, the patient´s narrative was considered a critical component. Second, shared decision-making built on the partnership. Third, documentation on the digital platform contributed to the continuity and transparency of the partnership.

## Sampling and participants

The sample was recruited among the participants of the PROTECT project intervention group from June 2018 to January 2019. Recruitment, in general, took place within six weeks after completion of the intervention. The participants were purposefully selected according to initial sampling criteria (sex, age, educational level, civil status, number of phone calls and digital platform use) to capture the heterogeneity of factors influencing patient experience and intervention use in the studied sample (Table 2 shows the participant characteristics). Participants were enrolled until theoretical saturation was reached [20]. In total, 12 patients were invited to participate, and all agreed to participate.

Table 2. Participant characteristics (n = 12).

| Sex | |
|---|---|
| Female | 5 |
| Male | 7 |
| **Age** (years), mean | 71.4 |
| Median (range) | 73 (57–81) |
| **Diagnose** | |
| CHF* | 1 |
| COPD** | 9 |
| CHF and COPD | 2 |
| **Civil status** | |
| Living alone | 2 |
| Married/partner | 10 |
| **Education level** | |
| Compulsory | 4 |
| Secondary school | 5 |
| University | 3 |
| **Users of digital platform functions** | |
| Wrote own health plan | 4 |
| Personnel-documented health plan | 8 |
| Used self-ratings | 11 |
| Wrote messages | 6 |
| Invited family and friends | 2 |
| **Number of telephone calls** | |
| Median (range) | 3 (2–5) |
| **Total length of telephone calls** (min) | |
| Mean | 70 |
| Median (range) | 62 (35–124) |

* CHF, Chronic Heart failure.

** COPD, Chronic Obstructive Pulmonary Disease.

## Data collection

Face-to-face (n = 5) or telephone (n = 7) interviews were conducted to elicit data about patients' experience of the intervention. An interview guide S1 and S2 Figs contained the initial question, *'Can you please tell me about your thoughts when you found out you were offered person-centred telephone calls and access to a digital platform?'*. The guide also contained topic areas addressing the content and design of the PROTECT intervention, the care experience and patient reflections about the development of future digital health services. Additionally, probes and intermediate questions were used to clarify and encourage narration [20]. The interview guide was further elaborated, consistent with theoretical sampling, as the interviews proceeded [23]—the range of topics was narrowed after the tenth interview, to gather specific data to fill conceptual gaps or to answer analytical questions [20]. The interviews lasted 45 min on average (range 31–79 min) and were tape-recorded and transcribed verbatim.

## Analysis

Data collection and analysis, including initial and focused coding, constant comparison and memo writing, were conducted concurrently as described by Charmaz [20]. During initial coding each line was coded 'close to the data', with openness to explore theoretical possibilities. Later, focused coding was applied to synthesise and explain segments of data, using conceptual codes. Throughout the process, codes within and between interviews were systematically compared and sorted into categories [20]. Memo writing supported this iterative process, recording occurrences during data collection, analytical thoughts and ideas. Finally, analysis of the categories led to the core category. To ensure methodological rigour the authors assumed a reflexive stance in analysis; additionally, all authors participated in discussions about category development [20]. NVivo 11 software was used to organise the data during the analysis.

## Patient and public involvement

In the PROTECT project, patient and family representatives participated in discussions about the design of the intervention and of the study in a participatory process [24]. In the present study, a patient partner was recruited from the Swedish Heart and Lung Association (Riksförbundet Hjärtlung) and participated in discussions about the research question and the development of the interview guide. The patient partner also provided advice regarding the burden (time required) of participation in the study.

## Ethical approval and informed consent

The study was conducted in accordance with the principles of the Declaration of Helsinki and was approved (approval number: DNr 063–17) by the local Regional Ethical Review Board, Gothenburg, Sweden. All personal identifiers were removed from the data or disguised to preserve patient confidentiality. All participants provided written, informed consent.

# Results

## Being welcomed through the side door when lacking the front door keys

Patients used the digital PCC intervention to access informal interactions with healthcare professional when desired, to support self-management processes in everyday life. This experience was expressed in the core category *Being welcomed through the side door when lacking the front door keys*. This category can be understood as a supportive process initiated when patients perceived their customary strategies for maintaining health (on their own or in consultation with healthcare professionals) to be insufficient (i.e. lacking the front door keys). Underpinning the

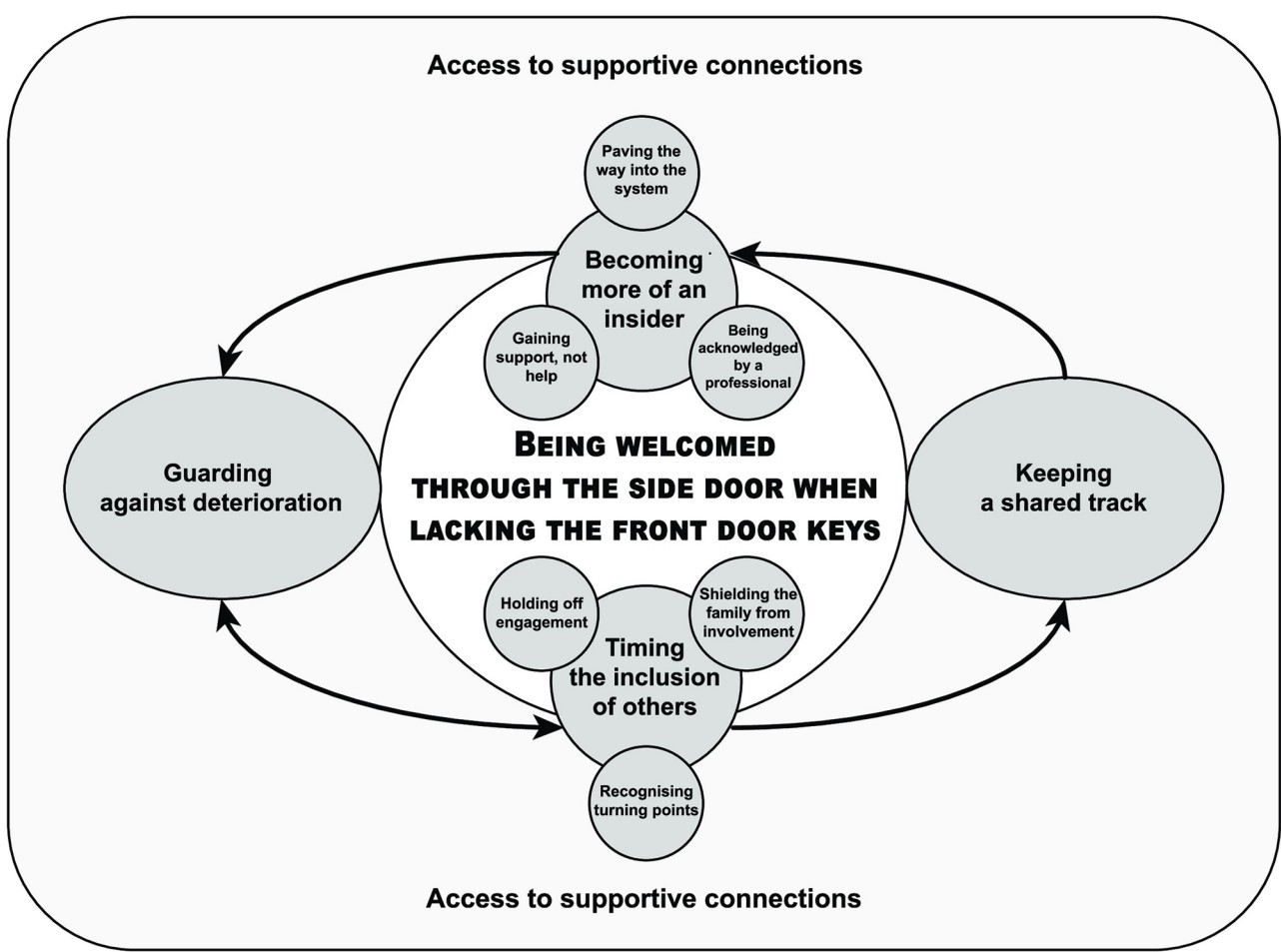

**Fig 1. The core category, categories and subcategories describing patients' experience of the e-health intervention.**

core category were five interrelated categories and six subcategories (Fig 1): The supportive process was first evident in the category *Guarding against deterioration* concerned with self-management actions in everyday life. A second category, *Keeping a shared track*, concerned interactions with professionals during the PCC intervention and how working in such partnership informed additional promotive or preventive actions. The patient's choice to work in partnership is described in a third category, *Timing the involvement of others*, and its subcategories, *Holding off engagement*, *Recognising turning points* and *Shielding the family from involvement*. The category *Becoming more of an insider* and its subcategories, *Paving the way into the system*, *Being acknowledged by a professional* and *Gaining support, not help*, concerned the experienced benefits of interactions during the PCC intervention. The final category, *Access to supportive connections*, described how interactions with intervention content and design were continuously shaped by personal, organisational, social and even interventional conditions.

**Guarding against deterioration.** The category *Guarding against deterioration* represents the ongoing process of protecting against the threat of health decline and loss of independence in daily life. Preventive actions (e.g., self-management strategies, adaptations and prioritisation of everyday activities) and the defence mechanism of avoiding thoughts of illness-related

health decline are included in this process. Patients initiated preventive actions by paying attention to their feelings and by being vigilant to changes in bodily signs/symptoms that could be signalling future impairment. One man expressed how the intervention supported his process of guarding against deterioration by giving him confidence in his ability to maintain health and independence:

> *What you are most afraid of is when you get something [COPD] and you fear you are rapidly deteriorating [at home], and then you will do even less. It feels as though I have gained some hope that this [decline in health] will not happen so fast.*

**Keeping a shared track.** The category *Keeping a shared track* illustrates that the intervention provided patients the possibility of merging their own expertise with that of an expert, to better understand the situation and available promotive and preventive options. For some patients, the intervention fostered self-awareness related to the chronic condition. Others discovered that daily rating of symptoms and wellbeing helped them recognise maintenance or changes in their health status, especially improvements. The telephone calls and personal health plans also contributed to *Keeping a shared track*, by helping patients to recognise and achieve important goals. One man described the benefit of exchanging and validating his ideas with a professional:

> *It is always good to have someone to discuss and listen to your difficulties. That's why people have [a need for] a sounding board to bounce ideas off. You have small ideas and you get evidence for these ideas, and that's how it works. That is the advantage of these phone calls.*

**Timing the involvement of others.** The category *Timing the involvement of others* reflects how participants' desire to work in partnership with professionals and family through e-health changed over time. Participants' interest in joining the intervention or engaging others when striving to protect their health, were reflected in three subcategories; *Holding off engagement, Recognising turning points* and *Shielding the family from involvement.*

**Holding off engagement.** *Holding off engagement* refers to a rejection of the intervention when patients felt either healthy (i.e., not in need of support) or too ill to look for help despite the recognised benefit of the intervention. This rejection was characterised by a cautious approach, display of little motivation or a low level of energy. Patients described that the timing of the intervention influenced whether the intervention was experienced to be of value. Nevertheless, understanding the continued availability of support once needed led to feelings of security even for patients choosing not to become fully engaged for the moment: '*Now I see my illness has improved since I started on a new medicine this autumn. But it feels secure to have this [support] if something happens.*'

**Recognising turning points.** *Recognising turning points* highlights the dynamic process of switching between self-management and working in partnership with the care provider during the intervention. This process was influenced by patients' perceptions of changes in their abilities to protect their health through self-management, in turn influencing the desire for support from others. A desire to work with a professional materialised with the recognition that something was changed and with subsequent recognition of knowledge gaps and the need for professional advice regarding interpretation and management of symptoms, i.e., lacking 'keys' to protect health. In contrast, gaining these keys during the intervention supported self-management:

> *After all, I had a lot of difficulty with my breathing, and it was tough. . ..without anyone to turn to, to get information to grasp [what is really going on]. Um, and there [on the digital*

*platform] you could rate your daily wellbeing, and you could ask questions. That would have been a huge help.*

**Shielding the family from involvement.**   *Shielding the family from involvement* highlights that patients attempted to shield family members from extra burden by not inviting them to participate on the digital platform. A number of patients considered everyday conversation to be more accessible for family members compared with use of the digital platform: '*I have not invited my husband as he would see it as just tagging along because I am the more tech-savvy in our family.*' The choice not to involve family in the partnership via the digital platform was influenced by several factors, including the need to preserve personal integrity, the desire to protect loved ones from worry and as described, the wish to avoid burdening them with yet another task. Later recognition of the possible benefits of integrating family into a partnership was associated with the development of more complex health needs with advancing age.

**Becoming more of an insider.**   The category *Becoming more of an insider* highlights patients' experience of being welcomed and encouraged to take responsibility for their health during and after the intervention. Patients' described that the PCC intervention strengthened their position in the healthcare system, by empowering them to determine the moment of shift between *self-management and working with a professional.* Three subcategories were identified to describe this: *Paving the way into the system*, *Being acknowledged by a professional* and *Gaining support, not help*.

**Paving the way into the system.**   *Paving the way into the system* describes patients' interactions with the content and design of the intervention, and the function of the intervention in providing a venue for rapid access to consultation with health professionals.

*I don't have to sit and wait on the telephone for hours on end. Instead, I could write [in the 'chat'], and they answered quickly. Maybe not in an hour, but an answer usually came the same day if I hadn't written too late in the afternoon.*

The ability to participate in the health system in this way was advantageous in that it afforded an opportunity for follow-up from healthcare experts to complement the patient's own expertise and self-checks/controls. The support, however, did not bridge barriers to usual care in instances where the links between the professionals in the e-health intervention and other care contacts were ambiguous—in these cases, remaining barriers led to feelings of disappointment and concern that the support had not paved the way to needed direct care.

**Being acknowledged by a professional.**   *Being acknowledged by a professional* describes patients' experience of health professionals' actions and responses, and resulting affirmative meetings, during the intervention. This was characterised by professional actions demonstrating both willingness to give time to the patient and an interest and valuing of the patient contribution to the dialogue. The intervention also confirmed that professionals recognised patients' existence outside of the clinical setting. All of these led to the sense of *being welcomed*. One woman expressed:

*They show interest and engagement, making it easier to open up and tell [my story]. If I feel that someone isn't that interested, then why should I share anything? I have often walked away from the primary care centre and thought, 'Why did I go there?' They didn´t show any interest, but here they did. They showed real interest. And then it becomes a lot easier to share things, too.*

The content and design of the intervention contributed in different ways to the patient experience of *Being acknowledged by a professional*. Telephone calls and interactions on the

digital platform conveyed that the professional caregiver had time for the patients—an unfamiliar experience for many patients. The telephone calls played a prominent role and contributed to patients' feelings of being cared for and belief in the of reliability of the support; additionally, the ability to participate in the formulation of the health plan further helped patients to feel worthy and acknowledged.

**Gaining support, not help.** *Gaining support, not help* highlights the reinforcement of participants' own skills. The health professionals encouraged patients to participate in their own care. At the start of the intervention, patients often expected a medical focus on the diagnosis of CHF or COPD; however, the telephone conversations addressed areas of concern in other ways (e.g., addressing management of social isolation and understanding of the influence of comorbidities and age-related health impairment on daily life). Additionally, the offer of support was distinguished from the offer of help by actions such as the extended invitation to the patient to write a health plan and the provision of guidance to the patient in building competencies in self-care (to promote health) or in navigating health services:

> *Well, maybe not really [getting help], but for advice on where to go. I have been given information about how I should proceed. I learned how to book appointments [via the web]. This was good for me.*

**Access to supportive connections.** The category *Access to supportive connections* entails experiences of having, or not having, the opportunity to come in contact with someone to lean on for emotional or practical support (inside or outside the PCC intervention). The category embodies two concepts: First, the category alludes to the existence of different levels of relationships promoting health within established healthcare and social networks. Second, the category denotes patients' perceived ability to use technology to access healthcare via e-health. Most participants expressed the feeling of being an outsider in the conventional healthcare system. The participants further described that access to mutual relationships in routine care varied over time, influenced by staff turnover, access to professionals with expertise in CHF or COPD and the responsiveness of professionals to requests for contact. In social networks, the number of contacts varied, as did their geographical and relational proximity, resulting in similarly varied patient experiences, with network relationships ranging from close to distant (lacking in everyday contact/support).

## Discussion

An important study finding was that the intervention allowed patients to choose their entry into partnership with health professionals. In contrast, the inclusion of family and friends as partners in the intervention held low priority for patients. These and the other findings shed light on the interactions that took place during the intervention and extend our understanding of how patient–professional partnerships can be realised remotely to support ongoing self-management processes in daily life. In addition, the findings help to explain when and why participation in the intervention was favourable. The patient experience was expressed through the core category, *Being welcomed through the side door when lacking the front door keys*, and its constituent categories and subcategories (Fig 1).

One interpretation of the core category is that the intervention provided a sense of security —participants could access a health professional by telephone or through the digital platform when needed. Furthermore, patients described access to mutual interactions, which conveyed acknowledgement of the patient as an equal partner. The importance of having access to genuine professional support augmenting patients' own capability to manage symptoms or

obstacles in daily life has been previously reported by patients diagnosed with COPD [25]; The finding is also consistent with those described in the TEN-HMS study [26] in which patients with a recent admission for heart failure were assigned systematic telephone support from specialist nurses. This (telephone) support can be compared with the remote measurement of weight, blood pressure and heart rate by automated devices linked to a single cardiology centre in the same study [26]. The number of admissions and mortality were similar in the two groups receiving different remote interventions, suggesting that reliable telephone access to a professional caregiver was equally effective in medical monitoring. In the present study, the telephone calls were experienced as a prominent element in the intervention, as illustrated by the subcategory *Being acknowledged by a professional*; however, our findings also highlight other promising strategies that demonstrate acknowledgement, including patient invitations to messages health professionals, to develop health plans or to conduct daily ratings (i.e., sharing in responsibility in promoting health). Our results indicate that during the intervention, these actions consolidated patients' experience of being recognised and respected and contributed to patients' experience of intervention interactions as person-centred.

Experiencing a sense of safety and security has repeatedly been reported to be beneficial in studies of the patient experience of e-health interventions [13,18,26]. Nevertheless, e-health can pose risks if patients' expectations of receiving care from professionals are not met—risk of disappointment and of delay in seeking direct care [13]. Interventions providing a flexible balance of self-management and access to professional support have been touted as a key to enhanced self-care among individuals with chronic conditions [27]. To our knowledge, this is the first study to demonstrate how, from a patient standpoint, e-health provided the opportunity for recurring shift between self-management and working in partnership. Such cooperation has been described as important in supporting self-efficacy [15,28] and as a requirement of supportive care for patients with COPD or CHF [28,29]. In our study, patients reported they protected their health by controlling the situation, i.e. *guarding against deterioration*, thus demonstrating self-management in line with previous research [27]. Similar to our finding, a meta-synthesis [13] of user experiences in telehealth (including remote monitoring in COPD) reported that such interventions brought welcomed responsibility and promoted self-care in patients. This strengthens our interpretation of the value of shared decision-making and documentation; however, in contrast to our findings, the risk for dependency and overtreatment has also been reported [13]. A possible explanation for this discrepancy is that normative structures in health care rely on the conventional roles of caregiver and care recipient, perpetuating the system of doing things for patients instead of integrating the patients as equal partners. Nevertheless, in our study, patients described that various interactive intervention functions supported self-management by providing the patients the opportunity, when needed, of *keeping a shared track*. As a result, patients could merge their expertise with professional competence to better understand the situation and available options and formulate a shared plan for how to move forward.

The category *Timing the involvement of others* deepens our understanding of why, when and to whom e-health should be offered. Our results show that patients engaged in professional partnership but, consistent with previous literature, preferred to protect family members from any adverse effects by *shielding them from involvement* [25]. Consistent with previous research, we found there was some resistance to joining the intervention when the skills required to use e-health support were incompatible with personal skills or when the support itself was seen to threaten personal identity and independence [19]. Flexible use of e-health services has been raised as a potential solution to increase intervention reach [30]. This argument is strengthened by our findings that flexibility in the choice of when to use the e-health service and for what purpose was experienced as beneficial in improving intervention reach.

Our results also demonstrate that the need for and motivation to be engaged in the intervention varied over time. To identify who would benefit most from the e-health intervention and under what circumstances, we recommend further studies investigating patterns of use of the various e-health support modalities employed in the PROTECT intervention, in larger samples.

Our findings indicate that the e-health intervention attracts individuals who lack mutual and sustainable relationships in conventional care settings. This observation is confirmed by studies showing that among individuals who were generally not considered to be capable partners during routine care (e.g., older people or adults without postsecondary education), those receiving PCC showed significantly more benefits than did those not receiving PCC [31,32]. Our findings identified how an e-health intervention can provide opportunities for person-centred interactions [33]. Patient experiences describing how the intervention contributed such opportunities were highlighted in the category *Becoming more of an insider* and its sub-categories. The subcategories *Being acknowledged by a professional* and *Gaining support, not help* illustrate that partnerships were facilitated by the creation of space for reliable and trust-worthy contact and by professionals who acknowledged patients as capable co-creators of care with shared responsibility. These findings agree with the person-centred processes that have elsewhere been described as central to different approaches translating PCC into practice [10,33,34]. Our results showed person-centred processes were successfully implemented in the e-health context and how the intervention contributed to these processes, suggesting that interactive e-health can be used to facilitate PCC. We believe that such support is useful in encouraging a preventive approach in primary care services. Overall, the intervention was seen as *Paving the way into the system*; however, this subcategory also reflected interprofessional and organisational barriers to PCC [33], underlining the need to facilitate collaboration throughout the care chain.

The implementation of PCC, in which partnership is essential, is challenging and highly influenced by contextual factors [12,33]. Our study not only adds to our understanding of contextual barriers to PCC implementation but also illuminates the opportunities for PCC in the e-health context. In our study, the category *Access to supportive connections* highlighted patients' experience of the context of care, describing requirements for sustainable relationships both during the intervention and in standard care. For example, the 'milieu' in the PROTECT intervention allowed patient's voice to be heard, while the positive response from professionals facilitated the partnership.

A grounded theory approach [20] was suitable in capturing the implemented interactions and contextual influences needed to understand patient experiences of the person-centred e-health intervention. All analyses are situated in time, place and culture and must be understood within the particular context of the study setting, the intervention and the studied sample [20]. That the intervention was conducted within a research context with a limited number of health professionals (as opposed to a 'real-life' healthcare setting with, for example, higher staff turnover) could have influenced the findings and should therefore be seen as a limitation of the study; however, the approach also allowed us to increase heterogeneity in the studied sample, as recommended [35], by including patients from units with different practice requirements. Based on previous recommendations, we also included patients diagnosed with COPD or CHF, as they have similar symptoms [28]—including both diagnoses supports the transferability of findings across patient groups. The inclusion of participants in a randomised controlled trial could have contributed to the higher number of participants eager to use e-health and thus may have affected the results. The eligible sample also limited the recruitment of people living alone. We attempted to meet these potential limitations by including a sample reflecting heterogeneity in sex, age, educational level and degree of use of the e-health

intervention. Theoretical sampling did not direct the inclusion of participants because of the aforementioned recruitment requirements but was used to fill gaps in category development [23].

## Conclusion

The PCC study intervention introduced to complement standard care in patients diagnosed with COPD or CHF shows promise as a strategy to increase patient access to and involvement in preventive care. The patients experienced the intervention as an effective measure to strengthen self-management processes. Moreover, the patient participants valued their partnership with health care professionals, although were reluctant to include family and friends as partners in the digital platform. Our findings suggest that e-health interventions with interactive components provide a strategy for initiating and sustaining patient–professional partnerships, and support timely preventive care processes.

## Supporting information

**S1 Fig. Interview guide in English.**
(PDF)

**S2 Fig. Interview guide in Swedish.**
(PDF)

## Acknowledgments

We want to thank the participants for sharing their experiences and the participating primary care centres for support and assistance in conducting the study. We are also grateful to patient research partner Eva Fredholm for constructive discussions throughout the research process and Dr Joanne Fuller and Professor Nicky Britten for valuable support during the translation process.

## Author Contributions

**Conceptualization:** Emmelie Barenfeld, Lilas Ali, Sara Wallström, Andreas Fors, Inger Ekman.

**Data curation:** Emmelie Barenfeld, Inger Ekman.

**Formal analysis:** Emmelie Barenfeld.

**Investigation:** Emmelie Barenfeld.

**Methodology:** Emmelie Barenfeld, Inger Ekman.

**Project administration:** Inger Ekman.

**Resources:** Inger Ekman.

**Validation:** Lilas Ali, Sara Wallström, Andreas Fors, Inger Ekman.

**Writing – original draft:** Emmelie Barenfeld.

**Writing – review & editing:** Lilas Ali, Sara Wallström, Andreas Fors, Inger Ekman.

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
