## [Decision Letter · Decision Letter 0]

28 Jul 2020

PONE-D-20-16256

Becoming more of an insider: A grounded theory study on patients’ experiences of a person-centred e-health intervention

PLOS ONE

Dear Dr. Barenfeld,

Thank you for submitting your manuscript to PLOS ONE. After careful consideration, we feel that it has merit but does not fully meet PLOS ONE’s publication criteria as it currently stands. Therefore, we invite you to submit a revised version of the manuscript that addresses the points raised during the review process.

It is recommended to face the following issues emerged during the review process (see for details the enclosed reviewers' comments):

- improuve readability correcting small errors with language and clarify the language in the title of Table 2

- provide further clarifications about the adopted sampling criteria

- simplify the language in category headings and in the accompanying text

- clarify whether and how the patients experienced their interactions with PCC as person-centred

- specify in describing the intervention, if there was a range of conversations offered

-clearify how the theoretical sampling has taken place

- provide details about recruitement of participants i.e. when were the patients asked to participate, how many were asked to participate, how many agreed

-better explain the context in some parts of the section "results"

- write "Standards for Reporting Qualitative Research" before the abbreviation “SRQR” (Page 4) .

We look forward to receiving your revised manuscript.

Kind regards,

Filomena Papa

Academic Editor

PLOS ONE

Journal Requirements:

2. Please include a copy of the interview guide used in the study, in both the original language and English, as Supporting Information, or include a citation if it has been published previously.

Reviewers' comments:

Reviewer's Responses to Questions

**Comments to the Author**

1. Is the manuscript technically sound, and do the data support the conclusions?

Reviewer #1: Yes

Reviewer #2: Yes

2. Has the statistical analysis been performed appropriately and rigorously? 

Reviewer #1: N/A

Reviewer #2: N/A

3. Have the authors made all data underlying the findings in their manuscript fully available?

Reviewer #1: No

Reviewer #2: No

4. Is the manuscript presented in an intelligible fashion and written in standard English?

Reviewer #1: Yes

Reviewer #2: Yes

5. Review Comments to the Author

Reviewer #1: This is a well written and important article which aims to explore the experiences of a person-centred e-health intervention in patients diagnosed with chronic obstructive pulmonary disease (COPD) or chronic heart failure (CHF).

Before considering publication, I have some questions which need to be clarified. Please see attachment.

Reviewer #2: This paper considers a novel intervention that seeks to support two important aspects of health service delivery:person-centred care and digitalisation. Specifically it considers the patient experience of participation in the intervention, an approach that is increasingly understood as a valuable way of investigating whether and how a new intervention works to support relevant outcomes.

Overall this study provides a good description of the intervention and the context for intervention delivery. In addition, the study is methodologically sound and the methods are well described. The study also highlights some interesting findings (e.g. at what points the patients find the intervention to be useful).

Below are a few suggestions for developing the paper further:

1. There are a number of small errors with language which compromise readability.

2. The paper provides a detailed list of participant characteristics /initial sampling criteria (Table 2). Again, the language in the title of the table is unclear. In addition, I think there needs to be further clarification as to which characteristics informed the sampling and why they were chosen.

3. The authors’ explanation of the core category and Fig 1 both provided useful ways of understanding the focus of the analysis. In addition, the study findings clearly lifted the data from a descriptive to theoretical perspective. However when reading the category headings (including the core category) and the accompanying text I often lost sense of how each category fitted into the bigger picture. I wonder if simplifying some of the headings and language may help make their relevance more accessible/meaningful to the reader?

4. I was really pleased to see the authors clearly define what made the intervention person-centred. However whilst the results emphasised that patients experienced the intervention as one that enhanced their ability to access healthcare professionals there seemed to be less focus on the patient experience of patient-centredness. I would have liked to come away with a clearer understanding of whether and how the patients experienced their interactions with HCPs as person-centred.

6. PLOS authors have the option to publish the peer review history of their article (what does this mean?). If published, this will include your full peer review and any attached files.

Reviewer #1: No

Reviewer #2: No

---

## [Author Response · Author response to Decision Letter 0]

5 Oct 2020

Thank you for the opportunity to revise our manuscript entitled Becoming more of an insider: A grounded theory study on patients’ experience of a person-centred e-health intervention. The comments from you and the reviewers have been very helpful when revising the manuscript. We would like to thank the reviewers for valuable and constructive comments. Your feedback and questions have helped to improve our revised version of the manuscript. Responses to the issues raised and our subsequent revisions are addressed below. 

Editor comments

Improve readability correcting small errors with language and clarify the language in the title of Table 2

On request the manuscript has been on a new round of language editing to improve readability, and these changes have been made accordingly throughout the document. The editing was performed by Edanz Group (https://en-author-services.edanzgroup.com/ac). We have edited the language in the title of table 2.

-provide further clarifications about the adopted sampling criteria

We have provided further clarifications about the adopted sampling criteria and updated the text (page 6, line 124-130).

-simplify the language in category headings and in the accompanying text

We have simplified the language in the accompanying text in the results section (page 9-16). We have also shortened the following category headings to improve readability; 

Being welcomed by the side door when lacking front door keys to protect health

Keeping a share track towards health

Having varying Access to supportive connections.

We have simplified the language in the accompanying text in the results section (page 9-16). We have also shortened the following category headings to improve readability; 

Being welcomed by the side door when lacking front door keys to protect health

Keeping a share track towards health

Having varying Access to supportive connections.

-clarify whether and how the patients experienced their interactions with PCC as person-centred

In this project we have not been able to study how the patients experienced their interactions with HCPs as person-centred specifically as our aim was to explore experiences of a person-centred intervention as a whole. However, we have clarified and extended the discussions of what can be interpreted as person-centred interactions from a patient perspective in the discussion (see line 347-348, line 356, 372-374, 395-399 and 422-426).

-provide details about recruitement of participants i.e. when were the patients asked to participate, how many were asked to participate, how many agreed

We have clarified when the patients were asked to participate, the number of invited participants and that all patients who were asked to participate agreed participation. (page 6, line 124-130)

-better explain the context in some parts of the section "results"

We have added information to provide context to the quotes as requested by reviewer 1. (page 10, line 205-207, page 11 line 220-221, page 14 line 285)

write "Standards for Reporting Qualitative Research" before the abbreviation “SRQR” (Page 4) .

We have added Standards for Reporting Qualitative Research (page 4, line 96)

- We have addressed an update of one additional source for funding in the cover letter.

We ensure that our manuscript meets PLOS ONE's style requirements.

Please include a copy of the interview guide used in the study, in both the original language and English, as Supporting Information, or include a citation if it has been published previously.

A copy of the interview guide in Swedish and English has been included as supporting information and referred to in the document. (Page 7 and 25)

a) If there are ethical or legal restrictions on sharing a de-identified data set, please explain them in detail (e.g., data contain potentially identifying or sensitive patient information) and who has imposed them (e.g., an ethics committee). Please also provide contact information for a data access committee, ethics committee, or other institutional body to which data requests may be sent

We have addressed point a) in data sharing statements in the cover letter.

Reviewer 1

This is a well written and important article which aims to explore the experiences of a person-centred e-health intervention in patients diagnosed with chronic obstructive pulmonary disease (COPD) or chronic heart failure (CHF).

Thank you

The number of phone calls was “determined based on an agreement between the patient and the health professionals”. Was there, however, a range of conversations offered?

We have clarified that one person-centered phone-call was offered as a minimum, and that there was no specified upper limit (page 5, line 103 and 106). We have also inserted a reference to our study-protocol which provides further details on the conversations in the intervention (page 5, line 112).

The table 1 is very informative concerning the platform and its use and accesses

Thank you

It is not quite clear how the theoretical sampling which is considered a major part of grounded theory has taken place. In the abstract you write that twelve patients were “purposefully sampled in accordance with the initial sampling criteria”. I read “the initial sampling criteria” as the sampling criteria used in the PROTECT study which is presented as an RCT study. Thus, patients were sampled among patients in the intervention group. Yet, sampling could still be made theoretical where constant comparison of the initial findings from the first sampled patients could inform the further sampling. I wonder whether this has taken place. On the other hand, at page 8 you write that “data collection and analysis were done concurrently”. Please clarify if theoretical sampling only has been done by as mentioned at page 7 “developing the interview guide in line with theoretical sampling. 

Thank you for a valuable comment. We have clarified that the initial sampling criteria was set for this study (page 6) and that theoretical sampling was performed by developing the interview guide (page 7, line 142-144). Due to the limited sample it was not possible to direct the sampling to specific individuals which has been added as a limitation in the method discussion (page 21, line 463-465).

Moreover, when were the patients asked to participate? How many were asked and did all agree?

We have clarified when the patients were asked to participate, the number of invited particpants and that all participants agreed. (page 6, line 124-130)

The grounded theory “Being welcomed through the side door when lacking the front door keys to protect health” is clearly developed and based on data in the form of citations. Sometimes the context needs to be better explained.

We have provided further details to provide contextual understanding (page 10, line 205-207, page 11 line 220-221, page 14 line 285)

For instance, the first quote at page 10, gained some hope that this [what does “this” refer to?] Please elaborate on the context or explain in the [ ].

We have elaborated on the context in the [ ], to clarify that this refers to health decline. (page 10, line 211)

The second quote at page 10. What does “ball blank” mean in the sentence “why people have a ball blank to bounce ideas off”?

We have replaced the word ball-blank with sounding-board. (page 11, line 224)

Overall the theory provides a deeper and meaningful understanding of the advantages experienced by patients having a remote intervention which is person-centered and at the same time e-based. Especially the flexibility in patients’ ability to choose when to be active and when NOT to be active in the intervention was very convincing.

Thank you

The figure gathers the theory well and is well placed at page 9.

Thank you

A well written discussion is provided which both discuss importance of the findings and acknowledges study limitations and how they were addressed. The implications of how the study can inform evaluation of the intervention in the overall study is also explained. Literature in the field is included satisfactorily.

Thank you.

Page 4 Standards for Reporting Qualitative Research should be written before the abbreviation “SRQR”.

We have added Standards for Reporting Qualitative Research (page 4, line 96)

Reviewer 2 

This paper considers a novel intervention that seeks to support two important aspects of health service delivery:person-centred care and digitalisation. Specifically it considers the patient experience of participation in the intervention, an approach that is increasingly understood as a valuable way of investigating whether and how a new intervention works to support relevant outcomes.

Overall this study provides a good description of the intervention and the context for intervention delivery. In addition, the study is methodologically sound and the methods are well described. The study also highlights some interesting findings (e.g. at what points the patients find the intervention to be useful).

Thank you.

1. There are a number of small errors with language which compromise readability.

To improve the readability in line with comment 1-3, the manuscript has been language edited by Edanz Group (https://en-author-services.edanzgroup.com/ac).

2. The paper provides a detailed list of participant characteristics /initial sampling criteria (Table 2). Again, the language in the title of the table is unclear. In addition, I think there needs to be further clarification as to which characteristics informed the sampling and why they were chosen.

We have edited the language in the table heading (page 7). Thanks to your comments we noticed that some of our intitial sampling criteria was not mention in the text and thus differed from the content of table 2. We have elaborated our text and clarified our intitial sampling criteria. (page 6, line 126-127).

3. The authors’ explanation of the core category and Fig 1 both provided useful ways of understanding the focus of the analysis. In addition, the study findings clearly lifted the data from a descriptive to theoretical perspective. However when reading the category headings (including the core category) and the accompanying text I often lost sense of how each category fitted into the bigger picture. I wonder if simplifying some of the headings and language may help make their relevance more accessible/meaningful to the reader?

Thank you for your positive comment and constructive feedback of how our findings could be presented to be more accessible to readers. Your comment made us aware that the whole picture of how the categories fit together needs to be clarified. We have rewritten the first paragraph in the results section to clarify this (page 9). We have also simplified the language in the text in the results-sections. In addition, we have shortened the name of the category headings when possible without loosing or changing the essence of data. (page 9-15)

4. I was really pleased to see the authors clearly define what made the intervention person-centred. However whilst the results emphasised that patients experienced the intervention as one that enhanced their ability to access healthcare professionals there seemed to be less focus on the patient experience of patient-centredness. I would have liked to come away with a clearer understanding of whether and how the patients experienced their interactions with HCPs as person-centred.

This is an interesting comment, and we agree that this is an important topic to study further. In this project we have not been able to study how the patients experienced their interactions with HCPs as person-centred specifically as our aim was to explore experiences of a person-centred intervention as a whole. We have added clarifications and deepened the results discussion of what we interprete as person-centred interactions (see line 347-348; 356; 371-374; 395-399 and 422-426).

---

## [Decision Letter · Decision Letter 1]

21 Oct 2020

Becoming more of an insider: A grounded theory study on patients’ experience of a person-centred e-health intervention

PONE-D-20-16256R1

Dear Dr. Barenfeld,

We’re pleased to inform you that your manuscript has been judged scientifically suitable for publication and will be formally accepted for publication once it meets all outstanding technical requirements.

Kind regards,

Filomena Papa

Academic Editor

PLOS ONE

Additional Editor Comments (optional):

Reviewers' comments:

Reviewer's Responses to Questions

**Comments to the Author**

1. If the authors have adequately addressed your comments raised in a previous round of review and you feel that this manuscript is now acceptable for publication, you may indicate that here to bypass the “Comments to the Author” section, enter your conflict of interest statement in the “Confidential to Editor” section, and submit your "Accept" recommendation.

Reviewer #1: All comments have been addressed

Reviewer #2: All comments have been addressed

2. Is the manuscript technically sound, and do the data support the conclusions?

Reviewer #1: Yes

Reviewer #2: Yes

3. Has the statistical analysis been performed appropriately and rigorously? 

Reviewer #1: N/A

Reviewer #2: N/A

4. Have the authors made all data underlying the findings in their manuscript fully available?

Reviewer #1: No

Reviewer #2: Yes

5. Is the manuscript presented in an intelligible fashion and written in standard English?

Reviewer #1: Yes

Reviewer #2: Yes

6. Review Comments to the Author

Reviewer #1: As mentioned, I perceive the changes made by the authors meet the comments made in a satistactory way.

Reviewer #2: I am happy with the revisions made by the authors to the original manuscript, and with the standard of English.

7. PLOS authors have the option to publish the peer review history of their article (what does this mean?). If published, this will include your full peer review and any attached files.

Reviewer #1: No

Reviewer #2: No

---

## [Editor Report · Acceptance letter]

10 Nov 2020

PONE-D-20-16256R1 

Becoming more of an insider: A grounded theory study on patients’ experience of a person-centred e-health intervention 

Dear Dr. Barenfeld:

I'm pleased to inform you that your manuscript has been deemed suitable for publication in PLOS ONE. Congratulations! Your manuscript is now with our production department. 

Kind regards, 

on behalf of

Dr. Filomena Papa 

Academic Editor

PLOS ONE